# Prevalence of keratoconus and keratoconus suspect, and their characteristics on corneal tomography in a population-based study

Hiroyuki Namba[1,2]*, Naoyuki Maeda[3], Hiroshi Utsunomiya[2], Yutaka Kaneko[2], Kenichi Ishizawa[4‡], Yoshiyuki Ueno[5‡], Koichi Nishitsuka[2,6]

1 Department of Ophthalmology, International University of Health and Welfare School of Medicine, Narita City, Chiba, Japan, 2 Department of Ophthalmology and Visual Sciences, Yamagata University Faculty of Medicine, Yamagata City, Yamagata, Japan, 3 Department of Ophthalmology, Osaka University Graduate School of Medicine, Suita City, Osaka, Japan, 4 Department of Neurology, Hematology, Metabolism, Endocrinology and Diabetology, Yamagata University Faculty of Medicine, Yamagata City, Yamagata, Japan, 5 Department of Gastroenterology, Yamagata University Faculty of Medicine, Yamagata City, Yamagata, Japan, 6 Department of Ophthalmology, Saitama Medical Center, Kawagoe City, Saitama, Japan

☯ These authors contributed equally to this work.
‡ KI and YU also contributed equally to this work.
* hnamba@iuhw.ac.jp

**Data Availability Statement:** Relevant data are within the paper are also available at: https://doi.org/10.6084/m9.figshare.26039557.

## Abstract

Keratoconus (KC) is a progressive corneal disorder resulting in severe visual impairment. We aimed to determine the prevalence and corneal tomographic characteristics of KC and keratoconus suspect (KCS) in a population-based study, and to construct discrimination models with or without corneal tomography. A total of 1,544 eyes (822 participants aged ≥35 years) were evaluated using data from the Yamagata Study (2015–2017). Systemic and ophthalmological examinations including corneal tomography with swept-source anterior segment optical coherence tomography (AS-OCT) were conducted to determine the prevalence and corneal tomographic characteristics of KC and KCS. In addition, data on 766 eyes were used to construct discrimination models with or without corneal tomography. In results, KC was diagnosed in six (0.85%) participants, and KCS was diagnosed in 27 (1.46%) participants. The values including corneal power, keratometric cylinder, corneal central and thinnest thickness, corneal asymmetry, higher-order irregularity, and their inter-eye differences were associated with KC and KCS. The areas under the receiver operating characteristic curves for the three multivariate discrimination models (without corneal tomography, with corneal tomography, and without corneal tomography + inter-eye difference models) for participants with KC or KCS were 0.848, 1.000, and 0.930, respectively. When corneal tomography is unavailable, inter-eye differences in corneal parameters may be useful screening tools for KC and KCS.

## Introduction

Keratoconus (KC) is a progressive disorder characterized by stromal thinning and anterior corneal protrusion resulting in severe visual impairment due to corneal higher-order

**Funding:** The authors received no specific funding for this work.

**Competing interests:** The authors have declared that no competing interests exist.

aberrations (HOAs) and stromal scarring. KC typically appears of their adolescence, and its progression ceases in the fourth decade of life [1–3]. While conical protrusion of the cornea eventually affects both eyes, enlargement of the inter-eye difference (IED) or laterality, is a characteristic of KC [4–7].

While eyeglasses or contact lenses can provide useful vision corrections during the early stages of KC, screening for KC before performing LASIK, advanced surface ablation, phakic intraocular lenses, and refractive cataract surgery is critical for preventing postoperative keratectasia and / or deterioration of vision quality [8–11].

Corneal crosslinking has been suggested for individuals with progressive KC to slow or halt disease progression [12–14]. An early diagnosis is required to stabilize the disease at an earlier stage, thereby minimizing vision impairment and reducing the long-term effects on quality of life [15].

KC is typically diagnosed after corneal irregular astigmatism has become clinically significant. However, the development of the Placido-based corneal topographer [16], scanning [17], the Scheimpflug-based corneal tomographer [18–20], and anterior segment optical coherence tomography (AS-OCT) [21–23] have enabled clinicians to detect not only clinical KC, but also keratoconus suspect (KCS), and forme fruste KC [17, 24–27]. However, corneal tomography using Scheimpflug camera or AS-OCT is not always available in the clinic.

The reported prevalence of KC varies with a range of environmental, genetic, ethnic factors and the instruments, diagnostic criteria, and study designs [28–39]. For example, Kennedy et al. reported a disease prevalence of approximately 54.5 per 100,000 people in the United States based on conventional examinations [29], which differed from the reported prevalence in Israel (2.34%) [34] and Iran (0.76%) [18] where modern corneal topography/tomography was used.

Therefore, this study determined the prevalence of KC using corneal tomography, investigated the corneal tomographic characteristics of KC and KCS, and constructed predictive models of KC and KCS with or without corneal tomography.

## Materials and methods

### Participants

This study was conducted as part of the Yamagata Study, a population-based epidemiological investigation examining systemic and ophthalmologic disorders in Japanese individuals aged $\geq$ 35 years [40–44]. Present research was performed by the residual medical checkups in Funagata town which was divided into three areas: Funagata, Horiuchi, and Nagasawa areas in terms of schedule. Systemic and ophthalmic data included in this research were collected from residents of Funagata area on June 14 and 28, 2015; Horiuchi area on June 4 and 19, 2016; and Nagasawa area on June 4, 2017. Written informed consent was obtained from all study participants, and their personal information was isolated from the ophthalmological data by linkable anonymizing. This study adhered to the tenets of the Declaration of Helsinki and its later amendments. The Yamagata Study was approved by the Ethical Review Committee of the Yamagata University Faculty of Medicine (Yamagata, Japan). Person- and eye-specific investigations were performed on the participants. The participant selection process is presented in Figs 1 and 2. Because of their analogue, figure-based estimations, the tomographic patterns were investigated in eyes examined both by SS-1000 and by SS-2000. Nevertheless, the risk associations with KC/KCS were investigated only in eyes examined by SS-2000 to avoid numerical differences from the instruments.

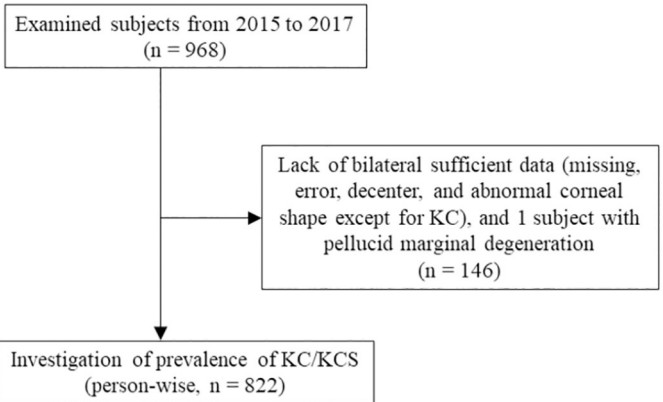

**Fig 1. Flowchart of the data-filtering procedure in person-specific investigations.** A total of 822 participants were included in this study.

## Examinations

Refractive spherical and cylindrical power, corneal cylindrical power, and intraocular pressure (IOP) were measured using an auto-ref/kerato/tonometer (TONOREF II, Nidek Co., Ltd., Aichi, Japan). Axial length was measured using a swept-source optical coherence tomography (OCT)-based biometer (OA-2000, TOMEY Corp., Aichi, Japan). Corneal tomography was performed through swept-source AS-OCT (CASIA SS-1000: 2015 examinations and CASIA2 SS-2000: 2016 and 2017 examinations, Tomey Corporation; Aichi, Japan) [44–46]. Physical characteristics such as height and weight were also recorded.

## Diagnosis of keratoconus and corneal tomographic analysis

KC and KCS were diagnosed by two experienced corneal specialists (HN and NM). KC was defined as eyes that showed visual acuity less than 20/20 and keratoconus pattern in axial power maps of the anterior corneal topography such as abnormal localized steepening and / or an asymmetric skewed bow-tie pattern. Additionally, the presence of abnormal elevation on the posterior surface and corneal thinning at the cone was confirmed (Fig 3). KCS was defined as eyes that had visual acuity 20/20 or better and keratoconus patterns in the anterior and posterior corneal surfaces (Fig 4) [3, 23–26]. Because there were few eyes with forme fruste keratoconus, which has irregularity only on the posterior surface, we did not include them.

Tomographic patterns were evaluated to analyze the corneal shapes among eyes with KC/KCS and control eyes, as described by Fuchihata et al. [47]. The axial power maps were classified into eight patterns: round, oval, symmetric bowtie, asymmetric bowtie, central steepening, lazy eight, inferior steepening, and crab claw. The definition of "central steepening" was K-readings > 47.0 diopters (D), which is outside the mean plus two standard deviations, in addition to the topographic pattern. Elevation maps of the anterior and posterior surfaces were categorized into nine patterns: central regular ridge, central irregular ridge, central incomplete ridge, central island, asymmetric regular ridge, asymmetric irregular ridge, asymmetric incomplete ridge, asymmetric island, and unclassified. Pachymetric maps were classified into six patterns: central round, central oval, paracentral round, paracentral oval, decentered round, and decentered oval. The term "central" was used when the thinnest point was located within the central 2-mm-diameter zone of the cornea. The map was classified as paracentral when the

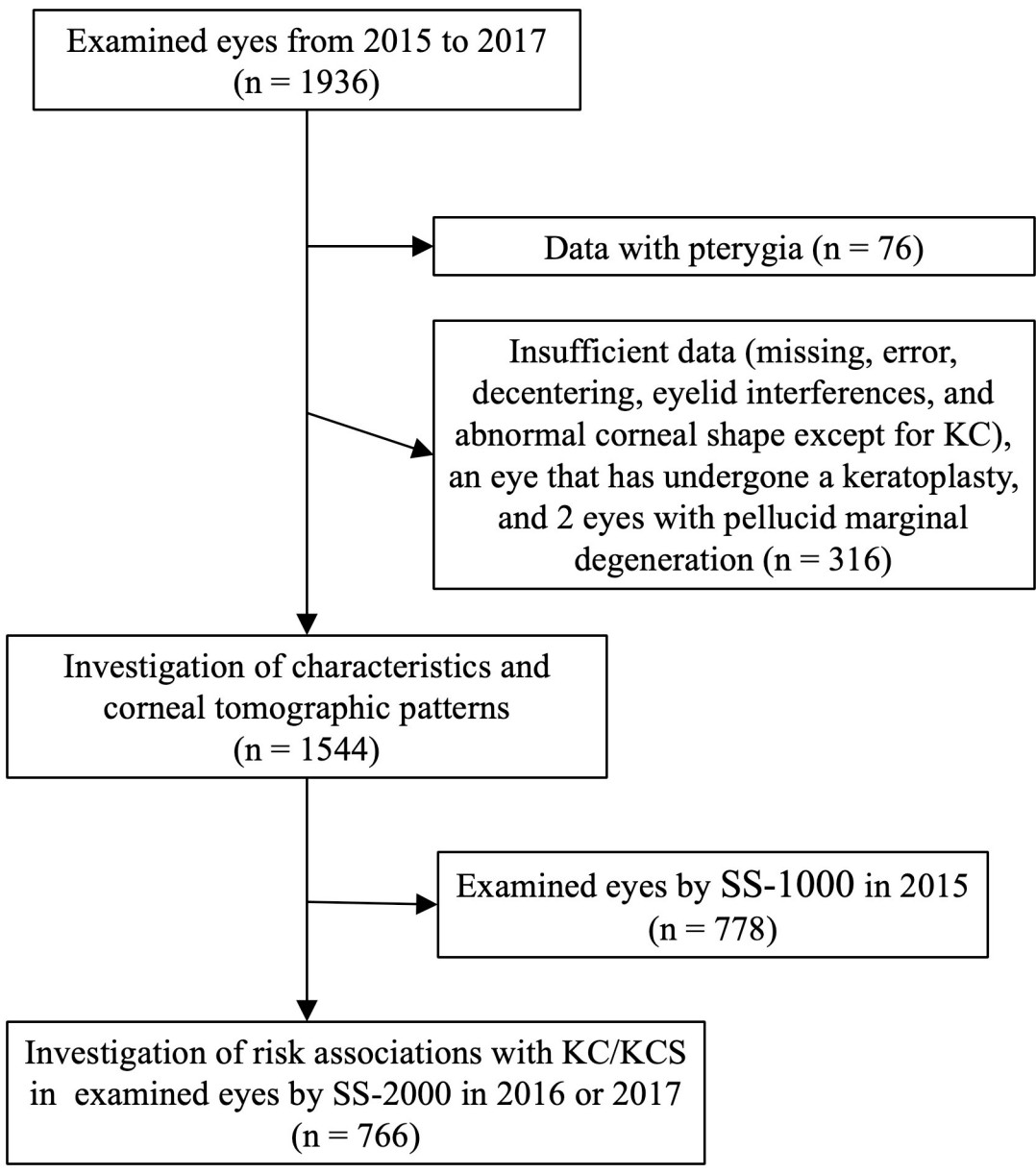

**Fig 2. Flowchart of the data-filtering procedure in eye-specific investigations.** A total of 1,544 eyes were included in the eye-specific analyses.

point was beyond the 2-mm-diameter zone and within a 3-mm-diameter zone. Similarly, the map was classified as decentered if the point was outside the 3-mm diameter zone.

Fourier analysis was performed using CASIA/CASIA2 software to evaluate corneal irregular astigmatism. Corneal dioptric data were expanded into spherical power, asymmetry components (first-order harmonic), regular astigmatism (second-order harmonic), and higher-order irregularity (HOI) components (third-and higher-order harmonics) within the central 6-mm zone were analyzed [48, 49].

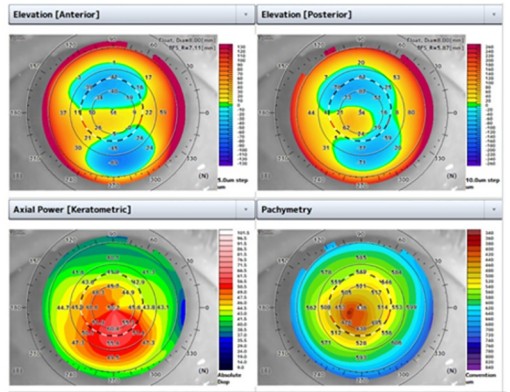

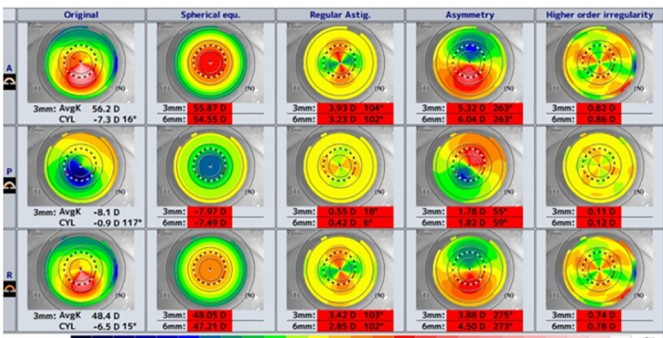

**Fig 3. A typical tomographic pattern of KC.** Abnormal localized steepening and elevation on the posterior surface, and corneal thinning were indicated.

## Statistical analyses

All statistical analyses were performed using STATA (version 14.2; StataCorp LLC, College Station, TX, USA) and SPSS (version 21.0; IBM Corp., Armonk, NY, USA) statistical software. Statistical significance was set at p <0.05. To compare person-specific and eye-specific characteristics, statistical tests were performed between non-KC, KCS, and KC groups. To determine the risk factors and discrimination models, non-KC and KC/KCS groups were included in statistical investigations. Clopper–Pearson exact confidence intervals (CIs) were evaluated to calculate 95% CIs for the prevalence of KC and KCS. A one-way analysis of variance and Tukey–Kramer tests were performed to compare person-specific characteristics within the non-KC, KCS, and KC groups. Kruskal–Wallis and Dunn–Bonferroni tests were used to estimate eye-specific characteristics. In corneal tomographic and pachymetric patterns investigations, the Chi-square test was used to compare distributions in the non-KC, KCS, and KC groups. Univariate logistic regression analyses were performed to determine the risk factors for KC and KCS. The odds ratios (ORs) and 95% CIs were calculated for each factor. The inter-eye differences (IEDs) in corneal parameters were determined by subtracting each variable's lower value from the higher value. The IEDs of the non-KC and KC/KCS groups were compared using the Mann–Whitney test. Multivariate discrimination models were established based on the identified risk factors for diagnosing KC/KCS. Receiver operating characteristic (ROC) curves of the discrimination models were drawn, and the area under the ROC curve (AUC) values were calculated to assess the sensitivity and specificity of the models. The cutoff point on the ROC curve was determined using the Youden index. Our data are available at: https://doi.org/10.6084/m9.figshare.26039557.

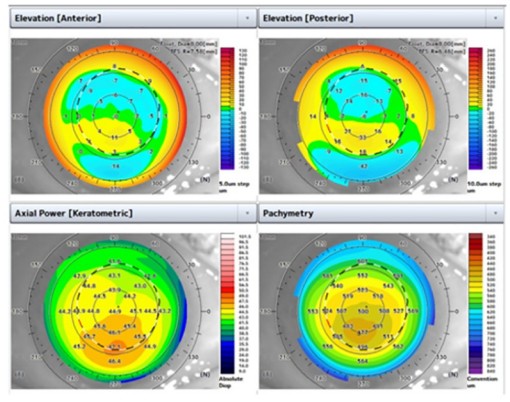

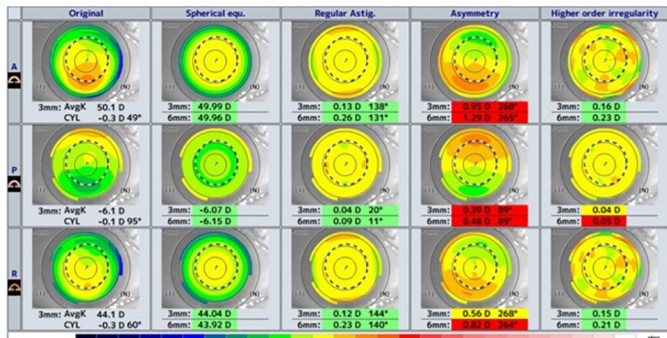

**Fig 4. A typical tomographic pattern of KCS.** Although the eye demonstrated similar pattern to KC, their visual acuity was 20/20.

## Results

### Demographic characteristics and prevalence of KC and KCS

A total of 822 participants (378 males and 444 females) were included in this study (Table 1). KC was detected in at least one eye in seven participants (0.85%, 95% CI: 0.41%–1.78%), and KCS was detected in at least one eye in 12 participants (1.46%, 95% CI: 0.82%–2.56%). The mean participant age was 62.3 ± 11.2 years, height was 159.1 ± 9.4 cm, and weight was 60.6 ± 11.5 kg (Table 1). Although the mean participant age had significant difference in the

**Table 1. Person-specific characteristics in non-KC, KCS, and KC subjects.**

| | Total | | non-KC | | KCS (at least one eye) | | KC (at least one eye) | | P values, one-way ANOVA | P values, Tukey–Kramer test | | |
|---|---|---|---|---|---|---|---|---|---|---|---|---|
| | Mean | SD | Mean | SD | Mean | SD | Mean | SD | | non-KC vs KCS | non-KC vs KC | KCS vs KC |
| Age (years) | 62.3 | 11.2 | 62.5 | 11.2 | 53.3 | 10.8 | 55.9 | 11.4 | 0.006 | 0.013 | 0.259 | 0.883 |
| Sex (male/female) | 378 / 444 | | 370 / 433 | | 5 / 7 | | 3 / 4 | | | | | |
| Height (cm) | 159.1 | 9.4 | 159.1 | 9.4 | 159.9 | 9.2 | 157.2 | 10.5 | 0.833 | 0.961 | 0.852 | 0.823 |
| Weight (kg) | 60.6 | 11.5 | 60.6 | 11.5 | 63.0 | 13.7 | 58.1 | 10.1 | 0.662 | 0.764 | 0.835 | 0.651 |
| Body mass index | 23.8 | 3.3 | 23.8 | 3.4 | 24.4 | 3.2 | 23.5 | 2.9 | 0.805 | 0.827 | 0.953 | 0.823 |

Abbreviation: KC = keratoconus; KCS = keratoconus suspect; SD = Standard Deviation; ANOVA = analysis of variance.

**Table 2. Eye-specific characteristics in non-KC, KCS, and KC subjects.**

| | Total (1544 eyes) | | non-KC (1515 eyes) | | KCS (20 eyes) | | KC (9 eyes) | | P values, Kruskal-Wallis test | P values, Dunn-Bonferroni test | | |
|---|---|---|---|---|---|---|---|---|---|---|---|---|
| | Mean | SD | Mean | SD | Mean | SD | Mean | SD | | non-KC vs KCS | non-KC vs KC | KCS vs KC |
| BCVA (logMAR) | 0.05 | 0.15 | 0.05 | 0.15 | -0.01 | 0.03 | 0.17 | 0.11 | < 0.001 | 0.082 | 0.001 | < 0.001 |
| IOP (mmHg) | 13.1 | 2.7 | 13.2 | 2.7 | 12.0 | 2.5 | 11.1 | 1.4 | 0.001 | 0.196 | 0.005 | 0.473 |
| Axial Length (mm) | 23.82 | 1.34 | 23.82 | 1.32 | 23.46 | 1.54 | 24.34 | 2.67 | 0.319 | | | |
| CCT (mm) | 531.4 | 30.5 | 532.1 | 30.3 | 508.5 | 19.9 | 499.7 | 21.8 | < 0.001 | < 0.001 | < 0.001 | 1.000 |
| Spherical equivalent (D) | -1.14 | 2.64 | -1.13 | 2.63 | -0.90 | 2.51 | -3.39 | 3.26 | 0.088 | | | |
| Refractive cylinder (D) | 1.00 | 0.71 | 0.99 | 0.70 | 0.98 | 0.61 | 2.06 | 1.78 | 0.114 | | | |
| Corneal power (D) | 43.96 | 1.55 | 43.93 | 1.50 | 44.25 | 1.98 | 46.33 | 2.90 | < 0.001 | 0.319 | < 0.001 | 0.108 |
| Corneal cylinder (D) | 0.81 | 0.72 | 0.79 | 0.65 | 1.04 | 0.71 | 2.58 | 1.17 | < 0.001 | 0.235 | < 0.001 | 0.005 |

Abbreviation: KC = keratoconus; KCS = keratoconus suspect; SD = standard deviation; BCVA = best corrected visual acuity; IOP = intraocular pressure; CCT = central corneal thickness.

analysis of variance, it was not considered as a confounder in subsequent investigations. Because KC is a young-onset disease, KC may have already developed until 35 years.

## Corneal tomographic and pachymetric patterns

A total of 1,544 eyes were included in the eye-specific analysis of corneal tomographic patterns, and 392 eyes were excluded due to pterygia or insufficient data. The mean central corneal thickness (CCT) in the non-KC, KCS, and KC groups was 532.1, 508.5, and 499.7 μm, respectively (Table 2). The CCT was significantly different between the non-KC and KCS groups (p<0.001) and between the non-KC and KC groups (p<0.001). The mean IOP was lower in the KC group than in the non-KC group (p = 0.005). The mean corneal power was lower in the non-KC group than in the KC groups (p<0.001). The corneal cylinders were significantly different between non-KC and KC groups (p<0.001), and between KCS and KC groups (p = 0.005).

In the axial power map, the inferior steeping pattern was the most frequently observed in 77.8% of eyes with KC and 75.0% with KCS (Table 3). The round pattern was the most

**Table 3. Classification of eyes in the axial power map according to topographic pattern.**

| Tomographic pattern in axial power map | non-KC | | KCS | | KC | |
|---|---|---|---|---|---|---|
| | N | % | N | % | N | % |
| Round | 938 | 61.9 | | | | |
| Oval | 204 | 13.5 | | | | |
| Symmetric bowtie | 285 | 18.8 | | | | |
| Asymmetric bowtie | 59 | 3.9 | 4 | 20.0 | 1 | 11.1 |
| Central steepening | 29 | 1.9 | | | 1 | 11.1 |
| Lazy 8 figure | | | 1 | 5.0 | | |
| Inferior steepening | | | 15 | 75.0 | 7 | 77.8 |
| Crab claw | | | | | | |
| Total | 1515 | 100 | 20 | 100 | 9 | 100 |

Abbreviation: KC = keratoconus; KCS = keratoconus suspect; N = number.

**Table 4. Classification of eyes in the elevation map according to topographic pattern.**

| Tomographic pattern in the elevation map | Anterior | | | | | | Posterior | | | | | |
|---|---|---|---|---|---|---|---|---|---|---|---|---|
| | non-KC | | KCS | | KC | | non-KC | | KCS | | KC | |
| | N | % | N | % | N | % | N | % | N | % | N | % |
| Central regular ridge | 1227 | 81.0 | 7 | 35.0 | 1 | 11.1 | 862 | 56.9 | 6 | 30.0 | 1 | 11.1 |
| Central irregular ridge | 217 | 14.3 | | | | | 586 | 38.7 | 4 | 20.0 | | |
| Central incomplete ridge | 71 | 4.7 | 7 | 35.0 | 1 | 11.1 | 67 | 4.4 | 3 | 15.0 | 1 | 11.1 |
| Central island | | | | | 2 | 22.2 | | | | | 2 | 22.2 |
| Asymmetric regular ridge | | | 4 | 20.0 | 3 | 33.3 | | | 2 | 10.0 | 2 | 22.2 |
| Asymmetric irregular ridge | | | 2 | 10.0 | 1 | 11.1 | | | 5 | 25.0 | 2 | 22.2 |
| Asymmetric incomplete ridge | | | | | 1 | 11.1 | | | | | 1 | 11.1 |
| Asymmetric island | | | | | | | | | | | | |
| Total | 1515 | 100 | 20 | 100 | 9 | 100 | 1515 | 100 | 20 | 100 | 9 | 100 |

Abbreviation: KC = keratoconus; KCS = keratoconus suspect; N = number.

frequent pattern (61.9%) in the non-KC group. In both the anterior and posterior elevation maps, the most common pattern in eyes without KC was the central regular ridge (81.0% in the anterior elevation map; 56.9% in the posterior elevation map in non-KC group, and 35.0% in the anterior elevation map; 30.0% in the posterior elevation map in KCS group, Table 4). Conversely, the asymmetric regular ridge patterns were common in eyes with KC (33.3% in the anterior elevation map; 22.2% in the posterior elevation map). In the pachymetric map, the central patterns were the most common in the non-KC group (58.6%) (Table 5). Paracentral and decentered patterns were observed more frequently in eyes with KC and KCS than in non-KC eyes (p<0.001).

## Risk associations for KC and KCS

A total of 766 eyes were included in the eye-specific analysis of the risk associations with KC/ KCS. Some eye-specific factors were found to be associated with KC/KCS (Table 6). Corneal power (OR = 1.51; 95% CI = 1.11–2.05) and cylinder (OR = 2.86; 95% CI = 1.79–4.13) was positively correlated with KC/KCS diagnosis, whereas IOP (OR = 0.65; 95% CI = 0.49–0.86) and CCT (per 10 μm, OR = 0.61; 95% CI = 0.49–0.77) were negatively correlated with KC/KCS diagnosis. From the corneal tomography data, the posterior corneal cylinder (OR = 2.86; 95%

**Table 5. Classification of eyes in the pachymetric map according to topographic pattern.**

| Topographic pattern in the pachymetric map | non-KC | | KCS | | KC | |
|---|---|---|---|---|---|---|
| | N | % | N | % | N | % |
| Central round | 888 | 58.6 | 6 | 30.0 | 2 | 22.2 |
| Central oval | 197 | 13.0 | 3 | 15.0 | | |
| Paracentral round | 366 | 24.2 | 6 | 30.0 | 5 | 55.6 |
| Paracentral oval | 62 | 4.1 | 4 | 20.0 | 2 | 22.2 |
| Decentered round | 2 | 0.1 | 1 | 5.0 | | |
| Decentered oval | | | | | | |
| Total | 1515 | 100 | 20 | 100 | 9 | 100 |

Abbreviation: KC = keratoconus; KCS = keratoconus suspect; N = number.

**Table 6. Risk associations of characteristics with KC/KCS.**

| Variables | Mean | SD | Univariate logistic regression analysis | | |
| --- | --- | --- | --- | --- | --- |
| | | | Odds Ratio | 95% CI | P value |
| IOP (mmHg) | 12.8 | 2.7 | 0.65 | 0.49, 0.86 | 0.003 |
| Axial length (mm) | 23.8 | 1.3 | 1.34 | 0.94, 1.91 | 0.101 |
| CCT (μm) | 531.8 | 29.5 | 0.61 (per 10 μm) | 0.49, 0.77 | < 0.001 |
| TCT (μm) | 523.6 | 29.5 | 0.50 (per 10 μm) | 0.38, 0.65 | < 0.001 |
| Corneal power (D) | 43.86 | 1.48 | 1.51 | 1.11, 2.05 | 0.009 |
| Corneal cylinder (D) | 0.80 | 0.73 | 2.86 | 1.79, 4.56 | < 0.001 |
| Anterior asymmetry (D) | 0.35 | 0.32 | 16.55 (per 0.1 D) | 1.65, 166.41 | 0.017 |
| Anterior HOI (D) | 0.15 | 0.05 | 4.17 (per 0.1 D) | 1.89, 9.21 | < 0.001 |
| Posterior corneal power (D) | -6.23 | 0.25 | 0.72 (per 0.1 D) | 0.61, 0.84 | < 0.001 |
| Posterior corneal cylinder (D) | 0.28 | 0.12 | 2.26 (per 0.1 D) | 1.60, 3.19 | < 0.001 |
| Posterior asymmetry (D) | 0.08 | 0.12 | 49.97 (per 0.1 D) | 6.09, 409.73 | < 0.001 |
| Posterior HOI (D) | 0.03 | 0.01 | 3.63 (per 0.01 D) | 2.34, 5.62 | < 0.001 |

Abbreviation: SD = standard deviation; CI = confidence interval; D = diopter; IOP = intraocular pressure; CCT = central corneal thickness; TCT = thinnest corneal thickness; HOI = higher-order irregularity.

CI = 1.79–4.56), anterior asymmetry (per 0.1 D, OR = 16.55; 95% CI = 1.65–166.41), posterior asymmetry (per 0.1 D, OR = 49.97; 95% CI = 6.09–409.73), anterior HOI (per 0.1 D, OR = 4.17; 95% CI = 1.89–9.21), and posterior HOI (per 0.01 D, OR = 3.63; 95% CI = 2.34–5.62) were positive risk factors, whereas the thinnest corneal thickness (TCT, per 10 μm, OR = 0.50; 95% CI = 0.38–0.65) and posterior corneal power (per 0.1 D, OR = 0.72; 95% CI = 0.61–0.84) were negative risk factors.

The IEDs for some eye-specific factors were also identified as risk factors for KC/KCS (Table 7). Specifically, the IEDs in corneal power (OR = 3.09; 95% CI = 1.76–5.42), cylinder (OR = 2.03; 95% CI = 1.40–2.94), axial length (OR = 6.23; 95% CI = 2.16–17.91), CCT (per

**Table 7. Risk associations of inter-eye differences with KC/KCS.**

| Variables | Mean | SD | Univariate logistic regression analysis for KC/KCS | | |
| --- | --- | --- | --- | --- | --- |
| | | | Odds Ratio | 95% CI | P value |
| IOP (mmHg) | 1.2 | 1.1 | 0.74 | 0.36, 1.56 | 0.431 |
| Axial length (mm) | 0.2 | 0.2 | 6.23 | 2.16, 17.91 | 0.001 |
| CCT (μm) | 5.5 | 5.3 | 3.69 (per 10 μm) | 1.63, 8.34 | 0.002 |
| TCT (μm) | 5.0 | 5.1 | 3.81 (per 10 μm) | 1.61, 9.01 | 0.002 |
| Corneal power (D) | 0.33 | 0.47 | 3.09 | 1.76, 5.42 | < 0.001 |
| Corneal cylinder (D) | 0.39 | 0.63 | 2.03 | 1.40, 2.94 | < 0.001 |
| Anterior asymmetry (D) | 0.18 | 0.71 | 1.03 (per 0.1 D) | 1.00, 1.05 | 0.019 |
| Anterior HOI (D) | 0.06 | 0.45 | 1.02 (per 0.1 D) | 0.96, 1.08 | 0.544 |
| Posterior corneal power (D) | 0.05 | 0.09 | 5.40 (per 0.1 D) | 2.72, 10.73 | < 0.001 |
| Posterior corneal cylinder (D) | 0.06 | 0.08 | 2.05 (per 0.1 D) | 1.40, 2.99 | < 0.001 |
| Posterior asymmetry (D) | 0.03 | 0.07 | 5.23 (per 0.1 D) | 1.97, 13.92 | 0.001 |
| Posterior HOI (D) | 0.01 | 0.04 | 1.03 (per 0.01 D) | 0.97, 1.09 | 0.385 |

Abbreviation: KC = keratoconus; KCS = keratoconus suspect; SD = standard deviation; CI = confidence interval; D = diopter; IOP = intraocular pressure; CCT = central corneal thickness; TCT = thinnest corneal thickness; HOI = higher-order irregularity.

**Table 8. Constructed discrimination model for predicting KC/KCS.**

| Model | Variables | AUC | P value | Youden index | Cut-off point | Sensitivity at cut off point | Specificity at cut off point |
|---|---|---|---|---|---|---|---|
| Non-OCT | Corneal cylinder (D) | 0.848 | < 0.001 | 0.71 | -4.23 | 0.92 | 0.79 |
| | CCT (μm) | | | | | | |
| OCT | TCT (μm) | 1.000 | | | | | |
| | Anteroir asymmetry (D) | | | | | | |
| | Posterior asymmetry (D) | | | | | | |
| Non-OCT+IED | Corneal cylinder (D) | 0.930 | < 0.001 | 0.79 | -4.20 | 0.89 | 0.90 |
| | CCT (μm) | | | | | | |
| | IED. Corneal power (D) | | | | | | |
| | IED. Axial length (mm) | | | | | | |
| | IED. CCT (μm) | | | | | | |

Abbreviation: CI = confidence interval; AUC = area under the curve; D = diopter; IOP = intraocular pressure; CCT = central corneal thickness; TCT = thinnest corneal thickness; IED = inter-eye difference.

10 μm, OR = 3.69; 95% CI = 1.63–8.34), TCT (per 10 μm, OR = 3.81; 95% CI = 1.61–9.01) were positively associated with KC/KCS diagnosis. Furthermore, posterior indices of corneal power (per 0.1 D, OR = 5.40; 95% CI = 2.72–10.73), cylinder (per 0.1 D, OR = 2.05; 95% CI = 1.40–2.94), and asymmetry (per 0.1 D, OR = 5.23; 95% CI = 1.97–13.92) were positively associated with KC/KCS diagnosis.

## Multivariate discrimination model for diagnosing KC/KCS

The non-OCT discrimination model was calculated using eye data not obtained from AS-OCT (without corneal tomography), while the OCT discrimination model was calculated using AS-OCT evaluation data (with corneal tomography). The non-OCT + IED discrimination model was calculated using IED data. The equations for these discrimination models are as follows:

Non-OCT model

$$\text{Discrimination score} = 16.01 + 0.958 \times \text{corneal cylinder(D)} - 0.0414 \times \text{CCT(μm)}$$

OCT model

$$\text{Discrimination score} = 4517.80 - 56.38 \times \text{TCT(μm)} + 23363.96$$
$$\times \text{anterior asymmetry(D)} + 19712.43 \times \text{posterior asymmetry(D)}$$

Non-OCT + IED model

$$\text{Discrimination score} = 17.15 + 0.82 \times \text{corneal cylinder(D)} - 0.048 \times \text{CCT(μm)} + 0.56$$
$$\times \text{IED. Corneal power(D)} + 2.21$$
$$\times \text{IED. Axial length(mm)} + 0.12 \times \text{IED. CCT(μm)}$$

The details of these models are described in Table 8 and Fig 5. The sensitivity and specificity of the non-OCT model were 92% and 79%, respectively. The AUC of the non-OCT model was 0.85. The OCT model had a sensitivity of 100% and a specificity of 100%. The AUC of the OCT model was 1.0. The sensitivity of the non-OCT + IED model was 89%, specificity was 90%, and AUC was 0.90. All models could discriminate eyes with KC effectively (n = 5 in eyes examined using SS-2000 in 2016 or 2017).

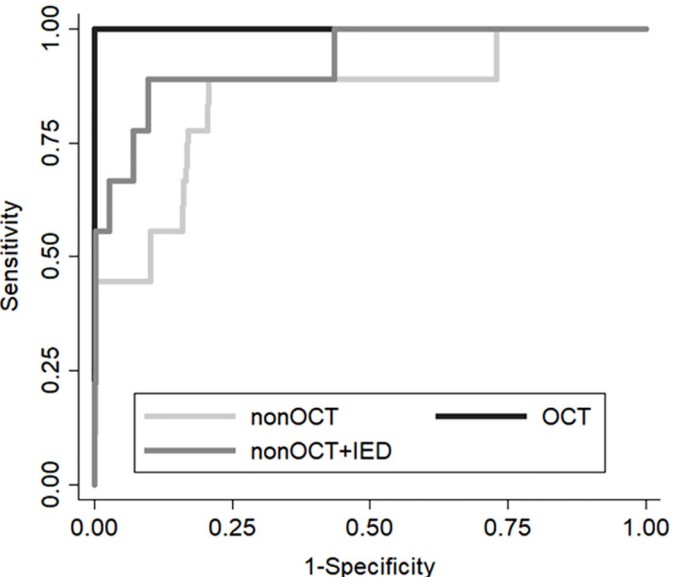

**Fig 5. Receiver operating characteristic curves of three discrimination models.** Non- OCT, OCT, and non-OCT + IED models were investigated.

## Discussion

This study provided the prevalence of KC (0.85%) and KCS (1.46%) and confirmed the tomographic and pachymetric patterns of the Japanese participants in each group. In addition, we investigated the risk factors for KC/KCS and constructed a discrimination model for predicting these diseases with and without corneal tomography.

The reported prevalence of KC varies greatly [28–39], one of which may be due to geographic differences. Pearson et al. reported that the prevalence of KC in Asian populations (229 per 100,000) is approximately four times that in Caucasian populations (57 per 100,000) [28]. The prevalence of KC in Middle Eastern, Egyptian, and Indian populations has been reported to be approximately 10- to 50-fold higher than that in European populations [18, 34–36]. Elbedewy et al. speculated that the high prevalence of KC in the Middle East might be associated with the high rate of chronic allergies and vernal keratoconjunctivitis in its populations [36]. In addition, continuous eye rubbing exerts a mechanical influence on the cornea, which may result in KC [50–52].

The differences in reported KC prevalence values may also be due to study design variations. Hospital-based or multicenter studies are typically retrospective, and diagnoses are based on data obtained through routine examinations [28–30]. Studies using registry data [31–23, 38] may have inclusion criteria similar to those of hospital-based studies conducted by ophthalmologists. These study designs likely exclude patients with early-stage KC who have not received medical consultation. Therefore, a population-based design may be better for detecting the actual prevalence of KC in the general population. Our study was population-based, and included relatively older subjects aged ≥ 35 years. This method may be effective to detect the actual prevalence because KC may have already developed until 35 years. If the younger subjects were included in the study, the surveyors may miss pre-disease stage patients.

Differences in examination instruments must also be considered when determining the prevalence of KC. Before the 1990s, KC was mainly diagnosed using slit lamp findings (such as Fleisher's ring, Vogt striae, and corneal thinning), keratometry, and retinoscopy. Therefore, it

was challenging to detect early-stage KC. The low KC prevalence in studies published during that period may be due to low diagnostic precision and low sensitivity [28–30, 39]. Afterward, Placido-based corneal topography enabled clinicians to detect irregular astigmatism of the anterior corneal surface in detail [16], and corneal tomographers based on slit-scanning or Scheimpflug cameras provided additional information regarding the cornea's posterior surface and thickness [8, 17, 20–27]. As a result, the prevalence of KC reported in studies using corneal topography or tomography [18, 34, 36] is higher than that obtained in previous studies due to the higher sensitivity of the instruments.

Few Japanese studies regarding the prevalence of KC have been reported. Tanabe et al. reported a very low prevalence of KC (9 per 100,000) [39]. This finding contradicts that reported in this study (796 per 100,000). However, this study was population-based, whereas the study by Tanabe et al. was hospital-based study. The results of hospital-based investigations may be biased toward moderate or severe cases of KC. While the previous studies did not include the diagnostic criteria or apparatus used in the diagnoses, corneal topography or tomography was not performed at the time of the study. In contrast, corneal tomography using AS-OCT, which allows detection of mild changes in tomographic abnormalities in the early stages of KC, was used in this study [22, 23].

The characteristics of tomographic maps of both eyes in KC or KCS or a control participant were analyzed in this study. In the axial power maps, the inferior steepening pattern was specific for KC and KCS (Table 3). In the anterior and elevation maps, the central regular ridge pattern was dominant in the non-KC (81.0%) and KCS groups (35.0%). Similarly, the central regular ridge pattern was dominant in the non-KC (56.9%) and KCS groups (30.0%) in the posterior elevation maps. In the pachymetric maps, the control participants demonstrated a central round pattern, and the paracentral patterns were dominant in the KC or KCS eyes. The population-based evaluation of the color map patterns of anterior and posterior corneal surfaces may be a unique feature of this study. It must be worth understanding the tomographic patterns of KC/KC for detecting them and also for the differential diagnosis in clinical practice.

In KC, sex difference has been discussed. While some studies have reported that the prevalence of KC is higher in male patients [18, 28, 30, 31, 33, 34], other studies have shown that the prevalence is not associated with sex [29, 36–38]. Jonas et al. reported an association between high-risk KC and the female sex [35]. In our study, no sex differences in prevalence were found in KC/KCS. The sex-specific difference in KC may remain controversial.

In the risk investigations of KC/KCS, the values including IOP, CCT, TCT, corneal power, cylinder, asymmetry, HOI, and posterior corneal indices, were associated with KC and KCS in this study, consistent with previous studies [3, 18, 36, 37].

In general, data of one eye of a patient were compared with data of affected and unaffected eyes of other individuals to avoid bias associated with a symmetry in laterality. However, patients with KC have obvious asymmetry in infero-superior differences in a single eye and IEDs. Zadnik et al. previously demonstrated IEDs in visual acuity and corneal cylinder [4], while Chopra reported differences in spherical error and the spherical equivalent [5]. In addition, Henriquez et al. and Dienes et al. found differences in the corneal cylinder, CCT, TCT, and posterior elevation of the cornea using Scheimpflug camera imaging [6, 7]. While these studies were all regarding KC, the data in this study suggest that IEDs in corneal indices are associated with both KC and KCS.

The diagnostic instruments used for the diagnosis of KC/KCS vary among studies. A discrimination model optimized for the study-specific purpose and environment is required for minimal and precise screening. In addition, model parameters should be generalizable and easily implemented in clinical settings. While the OCT model evaluated in this study yielded

the best precision for detecting KC and KCS (100% sensitivity and 96% specificity), AS-OCT is not widely used in clinics. In contrast, the addition of IED data improved the accuracy of the non-OCT model (from 90% to 95% sensitivity and 79% to 83% specificity), thereby increasing the AUC from 0.876 to 0.940. While eyes with KC in this study population were effectively discriminated in all models, non-OCT + IED is preferable for screening KCS when corneal tomography is unavailable.

A major strength of this study is that it is a population-based investigation utilizing corneal tomography performed by ophthalmologic clinical specialists. Hospital-based investigations are typically unfavorable for estimating disease prevalence. While reports using registry databases utilize highly populated (and frequently nationwide) data, these data represent accumulations of hospital-based data, the risk of missing KCS is probably high. Therefore, population-based examinations are better for investigating the real prevalence of KC/KCS. However, the instruments used in population-based study is usually limited. Diagnoses performed according to K-readings lack information about corneal tomography including corneal thinning, asymmetry, and corneal irregular astigmatism in KC and KCS. It is difficult to exclude examination errors or abnormal corneal shapes caused by other pathologies in the absence of visual inspections of color-coded maps by corneal specialists. In this regard, the present study has the advantage of being able to detect not only KC but also KCS with the aid of visual inspections of color coded-maps in a population-based study by using anterior segment OCT as a corneal tomographer. Although prediction models using deep learning or neural network methods have been progressing recently [53, 54], an exact diagnosis is needed for constructing learning models. The results of this study may also be useful for this purpose.

This study also has several limitations. First, the population size may have been small to investigate the prevalence of KC effectively. In addition, the CIs were imprecise; for example, the CI for KC prevalence ranged from 0.41 to 1.78%. Additional research conducted using a larger population is necessary. Second, the prevalence of KCS (1.46%), may have been underestimated due to the diagnostic criteria. Our data included a number of aged participants who might have had cataract or other diseases influencing visual acuity. Actually, some patients who had mild keratoconus patterns in axial power maps and visual acuities less than 20/20 were excluded from our analyses because of an "abnormal corneal shape." Third, participants aged < 35 years were not included in this study. This is an inherent limitation of the study design that only used examination data obtained from medical checkups for detecting adult diseases. However, the incidences of KC/KCS in young population may be underestimated, as cases that will progress to KC or KCS are not included. Therefore, the estimation of prevalence of KC/KCS in 35 years or older may be more effective for investigating the prevalence of KC or KCS. The high prevalence of KC indicated by this study's findings will help better understand and develop treatment strategies for KC/KCS in Japan. On the other hands, studies conducted in different populations or ethnicities may yield different results, and similar studies are required for various settings, populations, and ethnicities.

As recent advances in corneal crosslinking have induced a paradigm shift in treating KC, an early diagnosis is increasingly beneficial. In addition, the high incidence of KC in the general population suggests the importance of screening for KC before performing LASIK, placement of phakic IOL, and refractive cataract surgery [8–11, 55–57]. Toric, extended depth-of-focus, and multifocal IOL implantation should be avoided in keratoconic eyes to prevent suboptimal results due to corneal HOAs and postoperative refractive errors. If corneal topography or tomography is unavailable in the clinic, IEDs in corneal power, cylinder, and axial length can be used as screening tools for suspected KC.

## Supporting information

**S1 File. STROBE checklist v4 combined PlosMedicine.**
(DOCX)

## Acknowledgments

The authors thank the participants and the associate staff of the Yamagata Study (Funagata).

## Author Contributions

**Conceptualization:** Hiroyuki Namba, Naoyuki Maeda, Kenichi Ishizawa, Yoshiyuki Ueno, Koichi Nishitsuka.

**Data curation:** Hiroyuki Namba, Naoyuki Maeda, Hiroshi Utsunomiya, Yutaka Kaneko.

**Formal analysis:** Hiroyuki Namba.

**Investigation:** Hiroyuki Namba, Naoyuki Maeda, Yutaka Kaneko.

**Methodology:** Naoyuki Maeda.

**Project administration:** Kenichi Ishizawa, Yoshiyuki Ueno, Koichi Nishitsuka.

**Writing – original draft:** Hiroyuki Namba.

**Writing – review & editing:** Naoyuki Maeda, Koichi Nishitsuka.

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
