## [Decision Letter · Decision Letter 0]

11 Jun 2024

PONE-D-23-15898Prevalence of keratoconus and keratoconus suspect, and their characteristics on corneal tomography in a population-based studyPLOS ONE

Dear Dr. Namba,

Thank you for submitting your manuscript to PLOS ONE. After careful consideration, we feel that it has merit but does not fully meet PLOS ONE’s publication criteria as it currently stands. Therefore, we invite you to submit a revised version of the manuscript that addresses the points raised during the review process.

Please consider the literature mentioned by Reviewer 2 and address the issues about the age limitation mentioned by both reviewers. 

We look forward to receiving your revised manuscript.

Kind regards,

Timo Eppig

Academic Editor

PLOS ONE

Journal Requirements:

Reviewers' comments:

Reviewer's Responses to Questions

**Comments to the Author**

1. Is the manuscript technically sound, and do the data support the conclusions?

Reviewer #1: Yes

Reviewer #2: Yes

2. Has the statistical analysis been performed appropriately and rigorously? 

Reviewer #1: Yes

Reviewer #2: I Don't Know

3. Have the authors made all data underlying the findings in their manuscript fully available?

Reviewer #1: No

Reviewer #2: No

4. Is the manuscript presented in an intelligible fashion and written in standard English?

Reviewer #1: Yes

Reviewer #2: Yes

5. Review Comments to the Author

Reviewer #1: The authors aimed to determine the prevalence of keratoconus (KC), investigate the corneal tomographic features of KC and keratoconus suspect (KCS) using corneal tomography, and develop predictive models for KC and KCS based on corneal tomography, comparing those with and without corneal tomography. However, there are methodological errors in this study, and it needs improvement.

1. The identification of participants in this study is referenced to an article from 1999. However, it was stated that data from 2015-2016 and 2017 were obtained and evaluated after mentioning this. It is essential to provide a clear explanation of the population included in the study.

2. In the literature, there is a population-based keratoconus prevalence study using corneal tomography.*

---

## [Author Response · Author response to Decision Letter 0]

17 Jun 2024

The responses to the comments of the reviewers

Journal Requirements:

RESPONSE: 

Thank you for the comment, and we revised our manuscript to meet PLOS ONE's style requirements (title page, paragraphs, etc.).

RESPONSE: 

Thank you for your point out. The associations of sex differences with ocular parameters are not the focus of our research. Therefore, we deleted the sentence containing the phrase "data not shown". We revised our paper as below: 

(Lines 388- 392 in Revised Manuscript with Track Changes)

“In our study, no sex differences in prevalence were found in KC/KCS. The sex-specific difference in KC may remain controversial.”

Comments to the Author

3. Have the authors made all data underlying the findings in their manuscript fully available?

Reviewer #1: No

Reviewer #2: No

RESPONSE: 

My apologies, it seems that the file we uploaded was incomplete. So we revised and uploaded the files again, separating them into eye- and person-specific data. And we revised our paper as below: 

(Lines 193 - 195 in Revised Manuscript with Track Changes)

“Our data are available at: https://doi.org/10.6084/m9.figshare.26039557.”

5. Review Comments to the Author

Reviewer #1:

The authors aimed to determine the prevalence of keratoconus (KC), investigate the corneal tomographic features of KC and keratoconus suspect (KCS) using corneal tomography, and develop predictive models for KC and KCS based on corneal tomography, comparing those with and without corneal tomography. However, there are methodological errors in this study, and it needs improvement.

1. The identification of participants in this study is referenced to an article from 1999. However, it was stated that data from 2015-2016 and 2017 were obtained and evaluated after mentioning this. It is essential to provide a clear explanation of the population included in the study.

RESPONSE: 

Thank you for this point out, and I apologize for the confusing wording. The basic structure of the study, including the division of the health examination schedule and the method of consent, follows that of the Yamagata study (Funagata study), so the paper from 1999 is also included in the references. The subjects of this study and the data obtained were newly examined and obtained from 2015 to 2017. 

We revised our paper as below: 

(Lines 96 - 105 in Revised Manuscript with Track Changes)

“This study was conducted as part of the Yamagata Study, a population-based epidemiological investigation examining systemic and ophthalmologic disorders in Japanese individuals aged ≥ 35 years [40–44]. Present research was performed in accordance with the residual medical checkups in Funagata town which was divided into three areas: Funagata, Horiuchi, and Nagasawa areas in terms of schedule. Systemic and ophthalmic data included in this research were collected from residents of Funagata area on June 14 and 28, 2015; Horiuchi area on June 4 and 19, 2016; and Nagasawa area on June 4, 2017.”

2. In the literature, there is a population-based keratoconus prevalence study using corneal tomography.*

RESPONSE: 

Thank you for the comment, and I apologize for the confusing wording again. We are not thinking that this is the first population-based study using tomography, but that this may be the first paper that color map “patterns” of the anterior and posterior surface and corneal thickness from tomography have been evaluated in a population-based design. 

We revised our paper as below: 

(Lines 379 - 383 in Revised Manuscript with Track Changes)

“The population-based evaluation of the color map patterns of anterior and posterior corneal surfaces may be a unique feature of this study.”

---

## [Decision Letter · Decision Letter 1]

12 Jul 2024

PONE-D-23-15898R1Prevalence of keratoconus and keratoconus suspect, and their characteristics on corneal tomography in a population-based studyPLOS ONE

Dear Dr. Namba,

Thank you for submitting your manuscript to PLOS ONE. After careful consideration, we feel that it has merit but does not fully meet PLOS ONE’s publication criteria as it currently stands. Therefore, we invite you to submit a revised version of the manuscript that addresses the points raised during the review process.

Could you please consider the comments #3-#7 from reviewer one which were in the original revision but due to a technical error the comments have not been forwarded to the authors.

We look forward to receiving your revised manuscript.

Kind regards,

Timo Eppig

Academic Editor

PLOS ONE

Reviewers' comments:

Reviewer's Responses to Questions

**Comments to the Author**

1. If the authors have adequately addressed your comments raised in a previous round of review and you feel that this manuscript is now acceptable for publication, you may indicate that here to bypass the “Comments to the Author” section, enter your conflict of interest statement in the “Confidential to Editor” section, and submit your "Accept" recommendation.

Reviewer #1: (No Response)

2. Is the manuscript technically sound, and do the data support the conclusions?

Reviewer #1: Yes

3. Has the statistical analysis been performed appropriately and rigorously? 

Reviewer #1: Yes

4. Have the authors made all data underlying the findings in their manuscript fully available?

Reviewer #1: Yes

5. Is the manuscript presented in an intelligible fashion and written in standard English?

Reviewer #1: Yes

6. Review Comments to the Author

Reviewer #1: Although the authors have sufficiently provided feedback for the first two of my comments, my comments #3 - 7 have not been addressed.

3. Keratoconus suspect is defined as the presence of keratoconus patterns on the anterior axial power map. After the development of corneal tomography devices, it is accepted that keratoconus develops from the posterior surface of the cornea. In this case, keratoconus suspect should primarily involve changes in the posterior surface patterns. The current definition might have missed keratoconus-suspected patients.

4. The mean age of patients is considerably higher than the typical age of keratoconus diagnosis. Were the patients diagnosed with keratoconus in this study newly diagnosed cases who had not received a diagnosis before the study? The presence of patients who had undergone treatments like CXL is not specified in the study.

5. It is recommended to remove soft and rigid contact lenses at least 1 week before keratometric measurements. Were the patients' contact lenses removed before measurements?

6.As a result of the study, using axial length in keratoconus suspect cases is not a clinically appropriate approach.

7. Determining sensitivity and specificity values in a study with 822 participants, including 7 keratoconus and 12 keratoconus-suspected patients, will not provide objective results.

7. PLOS authors have the option to publish the peer review history of their article (what does this mean?). If published, this will include your full peer review and any attached files.

Reviewer #1: No

---

## [Author Response · Author response to Decision Letter 1]

15 Jul 2024

I'm sorry for the lack of responses to the comments #3-#7. We were unable to receive the comments due to a technical error.

Reviewer #1: 

3. Keratoconus suspect is defined as the presence of keratoconus patterns on the anterior axial power map. After the development of corneal tomography devices, it is accepted that keratoconus develops from the posterior surface of the cornea. In this case, keratoconus suspect should primarily involve changes in the posterior surface patterns. The current definition might have missed keratoconus-suspected patients

RESPONSE: 

Thank you for the comment. In our research, we involved changes in both the anterior and posterior surfaces of the cornea as KC and KCS. Cases with only posterior surface changes are defined as forme fruste keratoconus (FFK) and are not included in our study. The reason for this is that there were only a few cases of FFK in this study, and they were either KC or KCS. There were no cases with only posterior changes in both eyes. We revised our paper as below: 

(Lines 137-140 in Revised Manuscript with Track Changes)

“KCS was defined as eyes that had visual acuity 20/20 or better and keratoconus patterns in the anterior and posterior corneal surfaces (Fig 4) [3,31-34]. Because there were few eyes with forme fruste keratoconus, which has irregularity only on the posterior surface, we did not include them.”

4. The mean age of patients is considerably higher than the typical age of keratoconus diagnosis. Were the patients diagnosed with keratoconus in this study newly diagnosed cases who had not received a diagnosis before the study? The presence of patients who had undergone treatments like CXL is not specified in the study.

RESPONSE: 

Thank you for your comment. We included KC whether newly diagnosed or previously diagnosed. One eye that had previously undergone a keratoplasty was excluded as a case with insufficient data in the ocular examination. Eyes that had undergone CXL were not included. CXL is not yet widely used in rural areas of Japan. We revised our paper as below: 

(Lines 4- 9 in Figure 2)

“Insufficient data (missing, error, decentering, eyelid interferences, and abnormal corneal shape except for KC), an eye that has undergone a keratoplasty, and 2 eyes with pellucid marginal degeneration (n = 316)”

5. It is recommended to remove soft and rigid contact lenses at least 1 week before keratometric measurements. Were the patients' contact lenses removed before measurements?

RESPONSE: 

Thank you for your point out. It is correct and we would have done the same if we could have. However, due to the limitation of a joint medical checkup project with the Department of Internal Medicine, and because prior restrictions on contact lens use would lead to inconvenience in daily life, we requested it, but it was difficult to enforce it. 

6.As a result of the study, using axial length in keratoconus suspect cases is not a clinically appropriate approach.

RESPONSE: 

Thank you for your comment. You raise a valid point, and the axial length measurement may not be accurate in KC with corneal shape abnormalities. However, in the investigations of KC risk assessment, our goal is to use the data obtained from clinical examinations to make a diagnosis of KC. I think the accuracy of axial length measurement in KC may not be an essential issue. 

7. Determining sensitivity and specificity values in a study with 822 participants, including 7 keratoconus and 12 keratoconus-suspected patients, will not provide objective results.

RESPONSE: 

I think you have a valid point. I think the number of cases may not be sufficient to calculate an epidemiologically accurate prevalence. However, the prevalence of KC in Japan has not been adequately studied in the past, and it is relatively close to the prevalence in China also in East Asia, therefore, I think we were able to show some significance. We indicated the limitation on the line 432-433.

OPTIONAL: 

We unified the terms "person-wise" and "eye-wise" into "person-specific" and "eye-specific" as below: 

(Lines 206 in Revised Manuscript with Track Changes)

“Table 1. Person-specific characteristics in non-KC, KCS, and KC subjects.”

(Lines 222 in Revised Manuscript with Track Changes)

“Table 2. Eye-specific characteristics in non-KC, KCS, and KC subjects.”

---

## [Decision Letter · Decision Letter 2]

1 Aug 2024

Prevalence of keratoconus and keratoconus suspect, and their characteristics on corneal tomography in a population-based study

PONE-D-23-15898R2

Dear Dr. Namba,

We’re pleased to inform you that your manuscript has been judged scientifically suitable for publication and will be formally accepted for publication once it meets all outstanding technical requirements.

Kind regards,

Timo Eppig

Academic Editor

PLOS ONE

Additional Editor Comments (optional):

Reviewers' comments:

Reviewer's Responses to Questions

**Comments to the Author**

1. If the authors have adequately addressed your comments raised in a previous round of review and you feel that this manuscript is now acceptable for publication, you may indicate that here to bypass the “Comments to the Author” section, enter your conflict of interest statement in the “Confidential to Editor” section, and submit your "Accept" recommendation.

Reviewer #1: All comments have been addressed

2. Is the manuscript technically sound, and do the data support the conclusions?

Reviewer #1: Yes

3. Has the statistical analysis been performed appropriately and rigorously? 

Reviewer #1: Yes

4. Have the authors made all data underlying the findings in their manuscript fully available?

Reviewer #1: Yes

5. Is the manuscript presented in an intelligible fashion and written in standard English?

Reviewer #1: Yes

6. Review Comments to the Author

Reviewer #1: All previously raised issues have been appropiately addressed and the manuscript now reads suitable for publication.

7. PLOS authors have the option to publish the peer review history of their article (what does this mean?). If published, this will include your full peer review and any attached files.

Reviewer #1: **Yes: **Eray Atalay

---

## [Editor Report · Acceptance letter]

1 Dec 2024

PONE-D-23-15898R2 

PLOS ONE

Dear Dr. Namba, 

I'm pleased to inform you that your manuscript has been deemed suitable for publication in PLOS ONE. Congratulations! Your manuscript is now being handed over to our production team.

Kind regards, 

on behalf of

Prof. Dr. Timo Eppig 

Academic Editor

PLOS ONE